# Association between systemic immune-inflammation index and chronic kidney disease: A population-based study

Lin Li[1], Kunfei Chen[2], Chengping Wen[1], Xiaoqin Ma[1]*, Lin Huang[1]*

1 Key Laboratory of Chinese Medicine Rheumatology of Zhejiang Province, Zhejiang Chinese Medical University, Hangzhou, China, 2 Hangzhou TCM Hospital Affiliated to Zhejiang Chinese Medical University, Hangzhou, China

☯ These authors contributed equally to this work.
* huanglin@zcmu.edu.cn (LH); 874503647@qq.com (XM)

**Data Availability Statement:** The full set of NHANES data used for this investigation is accessible to the general public here: https://wwwn.cdc.gov/nchs/nhanes.

## Abstract

### Background

Systemic immune-inflammation index (SII) is a new indicator of inflammation, and chronic kidney disease (CKD) has a connection to inflammation. However, the relationship between SII and CKD is still unsure. The aim of this study was whether there is an association between SII and CKD in the adult US population.

### Methods

Data were from the National Health and Nutrition Examination Survey (NHANES) in 2003–2018, and multivariate logistic regression was used to explore the independent linear association between SII and CKD. Smoothing curves and threshold effect analyses were utilized to describe the nonlinear association between SII and CKD.

### Results

The analysis comprised 40,660 adults in total. After adjusting for a number of factors, we found a positive association between SII and CKD [1.06 (1.04, 1.07)]. In subgroup analysis and interaction tests, this positive correlation showed differences in the age, hypertension, and diabetes strata (p for interaction<0.05), but remained constant in the sex, BMI, abdominal obesity, smoking, and alcohol consumption strata. Smoothing curve fitting revealed a non-linear positive correlation between SII and CKD. Threshold analysis revealed a saturation effect of SII at the inflection point of 2100 (1,000 cells/μl). When SII < 2100 (1,000 cells/μl), SII was an independent risk element for CKD.

### Conclusions

In the adult US population, our study found a positive association between SII and CKD (inflection point: 2100). The SII can be considered a positive indicator to identify CKD promptly and guide therapy.

**Funding:** This research was funded by Professor Chengping Wen's Zhejiang Nature Science Foundation (LQ20H270006) and Dr Lin Huang's National Key R&D Program of China (No. 2018YFC1705500). The funders are involved in research design, data collection and analysis, publication decisions, and manuscript preparation. Chengping Wen: conceptualization, supervision, review manuscripts; Lin Huang: conceptualization, data curation, supervision, review & editing manuscripts.

**Competing interests:** The authors have declared that no competing interests exist.

## Introduction

Chronic kidney disease (CKD) is a chronic disease manifested by renal impairment, with high morbidity and mortality imposing a heavy burden on public health [1, 2]. With an estimated prevalence ranging from 11–13% worldwide, CKD is a growing concern. Inflammation is one of those main factors responsible for the onset of cancers [3, 4]. The identification of modifiable factors is essential for the prevention of CKD, the delay of target organ damage, and the retardation of disease progression.

Systemic inflammation is a routine part of regular blood tests, a variety of biochemical or hematological indicators and ratios to identify signs of inflammation. Among these markers is the Systemic Immune-Inflammation Index (SII), which provides an inflammation composite index of peripheral lymphocyte, neutrophil and platelet counts. This new marker has gained popularity as it reflects not only the local immune response but also the systemic inflammation [5–7]. The SII has been tied to the occurrence and propagation of numerous cancers and has proven to be a valuable prognostic factor in cancer patients. Growing tumors are detected and engaged by the immune system, and in recent years, the seventh hallmark of cancer has been recognized to be the immune response and inflammation [8, 9]. In addition to tumors, SII has predictive value for cardiovascular diseases, neurological disorders, metabolic diseases, respiratory diseases, rheumatic diseases, etc [10–16]. SII has been shown to be positively associated with a number of kidney-related diseases. For example, SII has been positively associated with severe acute kidney injury [17] and prognosis in patients with renal cell carcinoma [18].

The potential correlation between SII and CKD is yet to be elucidated. Thus, we carried out a population-based cross-sectional investigation for search for the link between the SII and CKD in adults who took part in the National Health and Nutrition Examination Survey (NHANES).

## Materials and methods

### Data sources

Our study's data came from NHANES, a database of surveys put together by the Centers for Disease Control and Prevention (CDC). Participants are selected using a special sampling method, and their health and nutrition status are assessed. These results allow us to gain insight into the health and nutritional situation of the US public. Ethics review committees approve all study procedures, which are conducted every two years. Informed consent was obtained from participants before participation in the study. Interviews, physical examinations, and tests on blood and urine samples are all components of the investigation of NHANES. The mobile testing laboratory performed both physical and laboratory exams, and the interviews took place at the participants' houses. Participants in our data were recruited and participated in data and sample collection by CDC from 2003–2018. And this cross-sectional study was conducted in March 2023, and the information about individual participants was not identifiable during or after data collection.

### Study population

For the survey, we pooled data from 8 cycle years 2003–2018 and removed 14,444 subjects with missing SII data, 6,028 subjects with missing CKD data and 19,180 subjects under 20 years of age from the eligible population of 80,312. The selected participants in this study totaled 40,660 individuals. The sample selection process is visualized in Fig 1.

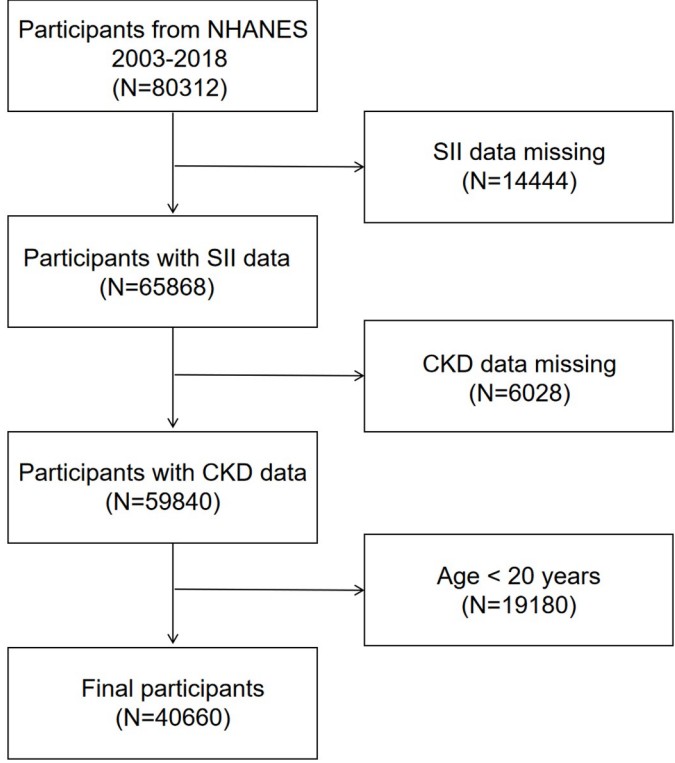

**Fig 1. Flowchart of participant selection.** NHANES, National Health and Nutrition Examination Survey.

## CKD

According to the Collaborative Epidemiology of Chronic Kidney Disease equation, CKD is indicated by estimated glomerular filtration rate (eGFR) < 60 mL/min/1.73m$^2$ or urine albumin-creatinine ratio (ACR) $\geq$ 30 mg/g [19]. Creatinine and urinary albumin were detected by Jaffe rate method and fluorometric immunoassay, respectively. The following is the eGFR calculation used by the CKD-Epidemiology Collaboration [20]:

$$
\begin{aligned}
\text{eGFR}_{\text{CKD-EPI}} &= 141 \times \min(\text{Scr}/\kappa, 1)^{\alpha} \\
&\times \max(\text{Scr}/\kappa, 1)^{-1:209} \times 0:993^{\text{Age}} \\
&\times 1:018[\text{if female}] \\
&\times 1:159[\text{if black}]
\end{aligned}
$$

Scr, which denoted serum creatinine concentration (mg/dL). The ACR was calculated using urine albumin/urine creatinine. Additionally, when the individual is female, $\kappa$ is 0.7 and $\alpha$ is -0.329, and when male, $\kappa$ is 0.9 and $\alpha$ is -0.411.

## SII

In this study, SII served as the independent variable, which was obtained by multiplying platelet count with the neutrophil count/lymphocyte count ratio. Using a CoulterDxH 800 analyzer, an automated hematology measurement instrument, these three blood cell types were examined and reported as 10$^3$ cells/mL.

## Covariates

In our study, covariates such as gender, age, race, marital status, education level, income poverty ratio, BMI, abdominal obesity, alcohol drinking, smoking, hypertension, and diabetes mellitus were considered. Race was divided into non-Hispanic white, non-Hispanic black, non-Hispanic Mexican-American, other Hispanic, and other races. Education level was classified as less than high school, high school or more than high school. Three divisions of the income-to-poverty ratio were established: 1.5, 1.5–3.5, and > 3.5. Marital status was classified into three groups: married/live with partner, widowed/divorced/separated, or never married. Drinking status was classified into 5 categories: never, former, mild, moderate, and heavy. For instance, individuals who had consumed < 12 drinks in their lifetime were defined as never drinkers, while former drinkers were those who had consumed $\geq$ 12 drinks in 1 year and did not drink last year. Mild, moderate, and heavy drinkers were defined based on the frequency and amount of alcohol consumption [21, 22]. Smoking status was Smoking status was divided into former, current, never smoker. People who had never smoked were defined as those who had never smoked $\geq$ 100 cigarettes in their lifetime, and those who had smoked $\geq$ 100 cigarettes but stopped smoking were defined as former smoker, and current smokers who were smoked intermittently or continuously smoked $\geq$ 100 cigarettes [23, 24]. Three BMI ranges were determined: below 25kg/m$^2$, 25–29.9 kg/m$^2$, and 30kg/m$^2$ and above. Abdominal obesity was defined as having a waist circumference $\geq$ 88 cm for women and $\geq$ 102 cm for males [25]. Hypertension was diagnosed if the mean systolic and diastolic were > 140 mmHg and/or 90 mmHg, respectively, after three measurements. Based on whether diabetic medications were taken, diabetes was diagnosed [26]. And the information about individual participants was not identifiable both during or after data collection.

## Statistical analysis

To evaluate the relationship between SII and CKD, we divided SII into quartiles varying from Q1 to Q4. The mean and standard deviation (SD) is used to describe a measure of continuous variables, whereas the percentage is used to describe a measure of categorical variables. Using weighted t-tests and weighted chi-square tests, we compared the differences between subjects grouped by SII quartiles and between subjects with or without CKD. To construct multivariate tests, we used multivariate logistic regression analysis between SII and CKD, with three models: Model 1 (without covariates), Model 2 (adjusted for age, sex, and race), and Model 3 (adjusted for all covariates). Odds ratio (OR) and 95% confidence interval (CI) were used in the models to assess SII and CKD. To perform sensitivity testing, we conducted subgroup analyses based on age, sex, BMI, abdominal obesity, alcohol consumption, smoking, hypertension, and diabetes. The nonlinear association and inflection point of SII with CKD were also investigated using smoothing curve fitting and threshold effect analysis models. R studio (version 4.2.2) and EmpowerStats (version 2.0) were used for all statistical analyses. *P* values of less than 0.05 were regarded as significant.

## Results

### Baseline characteristics of participants

Following the established criteria, 40,660 participants were involved in this study. Of these, 48.33% male and 51.67% female, with 49.77 ± 18.04 years on average. The ethnic breakdown was as follows: 20.66% Mexican American, 39.98% non-Hispanic white, 24.17% non-Hispanic black, 7.77% other Hispanic, and 7.42% other races. SII ± mean SD concentration was 550.60 ± 378.90 and 7447 patients with CKD, representing 18.32% of the total participants.

Table 1 shows that the presence of CKD showed statistically significant associations with various factors such as age, gender, race, education level, poverty, marital status, BMI, waist circumference, alcohol drinking and smoking status, hypertension, diabetes, and SII ($p < 0.05$). Compared to non-CKD, patients with CKD tend to be older, female, non-Hispanic white, less educated, widowed/divorced/separated, poor, BMI $>30 kg/m^2$, abdominal obesity, less heavy drink, more former smoking history, more hypertension, diabetes, and higher levels of SII.

Table 2 shows that SII quartiles were observed to have statistically significant ($p < 0.05$) correlates with age, gender, race, education level, poverty, marital status, BMI, waist circumference, alcohol and smoking status, hypertension, diabetes, and CKD. Compared to Q1, SII participants in the highest quartile Q4 tended to be older, female, non-Hispanic white, less educated, more widowed/divorced/separated, poor, with BMI $>30$ kg/m2, more abdominal obesity, more heavy drink, more former and current smokers, and more hypertension, diabetes and CKD patients.

## Association between SII and CKD

Since the effect values were not significant, SII/100 was used to magnify the effect values by a factor of 100. The outcomes obtained from the multivariate regression analysis of SII/100 and CKD are summarized in Table 3. The association was found to be significant in both the unadjusted Model 1 (1.06 (1.05, 1.06)) and the adjusted Model 2 (1.05 (1.04, 1.06)). Adjusted Model 3 showed a significant association of CKD with SII/100 (1.06 (1.04, 1.07)). Sensitivity analysis using SII quartiles is significantly positively correlated in model 1 (1.62 (1.51, 1.74)), model 2 (1.63 (1.51, 1.76)), and Q4 in model 3 (1.61 (1.41, 1.85)) compared to Q1. And for each unit increase in SII in subjects in Q4 compared to Q1, the risk of developing CKD increased by 61% (p for trend $< 0.05$).

We conducted further subgroup analyses to conduct sensitivity analysis. The findings proved that the association between SII and CKD exhibited statistically significant differences in the age, hypertension, and diabetes mellitus subgroups (Fig 2). However, no significant differences were observed in the gender, BMI, abdominal obesity, smoking, or drinking status subgroups. These findings indicate a significant impact of age, hypertension, and diabetes mellitus on the positive association between SII and CKD (*P* for interaction $< 0.05$). In addition, gender, BMI, abdominal obesity, smoking, and drinking status had no significant effect on the association (*P* for interaction$>0.05$).

In the age subgroup analysis, contrary to earlier findings, we discovered that SII had a negative association with CKD in the 30–39 age group [OR (95% CI): 0.97 (0.93, 1.02)]. However, the *P* values did not demonstrate statistical significance. And the positive association between SII and CKD was more pronounced above the age of 50, especially at 70–79. In addition, when stratified by hypertension or diabetes, our analysis indicated a connection between SII and CKD was more significant in hypertension and diabetes patients compared to those without hypertension and diabetes, respectively.

A smoothed curve fit was utilized to illustrate the nonlinear connection between SII and CKD (Fig 3). Through threshold analysis, it was discovered that SII had a saturation effect at the inflection point of 2100 (1,000 cells/μl). When SII $< 2100$ (1,000 cells/μl), SII and CKD were positively correlated, and when SII $> 2100$ (1,000 cells/μl), SII and CKD did not have a statistically distinct connection (Table 4). Therefore, according to this inflection point, it can be suggested that when we further explore the relationship between SII and CKD, it is more appropriate when SII is less than 2000 (1,000 cells/μl).

**Table 1. Weighted characteristics of the study population categorized by CKD status.**

| | Overall | No CKD | CKD | *p*-Value |
|---|---|---|---|---|
| | N = 40,660 | N = 33213 (81.68%) | N = 7447 (18.32%) | |
| Age (Mean ±SD, years) | 49.77 ± 18.04 | 45.09 ± 15.76 | 61.12 ± 17.37 | <0.0001 |
| Gender (%) | | | | <0.0001 |
| Male | 19653 (48.33%) | 48.94 | 42.78 | |
| Female | 21007 (51.67%) | 51.06 | 57.22 | |
| Race (%) | | | | <0.0001 |
| Mexican American | 8402 (20.66%) | 20.41 | 22.69 | |
| Non-Hispanic White | 16256 (39.98%) | 40.11 | 41.52 | |
| Non-Hispanic Black | 9826 (24.17%) | 23.7 | 24.44 | |
| Other Hispanic | 3161 (7.77%) | 8.39 | 5.3 | |
| Other Race | 3015 (7.42%) | 7.39 | 6.05 | |
| Education (%) | | | | <0.0001 |
| Less than high school | 10373 (25.54%) | 15.26 | 23.03 | |
| High school | 9391 (23.12%) | 23.22 | 26.32 | |
| More than high school | 20847 (51.33%) | 61.52 | 50.65 | |
| Marital status (%) | | | | <0.0001 |
| Never married | 7168 (17.64%) | 18.75 | 10.39 | |
| Maried/Living with Partner | 24383 (60.00%) | 64.9 | 57.55 | |
| Widowed/Divorced/Separated | 9087 (22.36%) | 16.35 | 32.06 | |
| Poverty ratio (%) | | | | <0.0001 |
| 0–1.5 | 16800 (45.14%) | 30.74 | 39.76 | |
| 1.5–3.5 | 9078 (24.39%) | 24.96 | 27.59 | |
| >3.5 | 11336 (30.46%) | 44.3 | 32.65 | |
| BMI (kg/m$^2$, %) | | | | <0.0001 |
| <25 | 11656 (29.06%) | 30.98 | 25.59 | |
| 25–29.9 | 13373 (33.35%) | 33.64 | 30.16 | |
| ≥30 | 15075 (37.59%) | 35.38 | 44.25 | |
| Abdominal obesity (%) | | | | <0.0001 |
| No | 22817 (58.97%) | 62.25 | 48.98 | |
| Yes | 15874 (41.03%) | 37.75 | 51.02 | |
| Drinking status (%) | | | | <0.0001 |
| Never | 5269 (14.54%) | 10.43 | 15.66 | |
| Former | 6258 (17.27%) | 12.57 | 23.06 | |
| Mild | 11979 (33.05%) | 36.13 | 35.84 | |
| Moderate | 5497 (15.17%) | 18.07 | 12.79 | |
| Heavy | 7241 (19.98%) | 22.8 | 12.65 | |
| Smoking status (%) | | | | <0.0001 |
| Never | 22277 (54.83%) | 55.19 | 50.51 | |
| Former | 9948 (24.48%) | 23.32 | 32.83 | |
| Now | 8406 (20.69%) | 21.49 | 16.66 | |
| Hypertension (%) | | | | <0.0001 |
| No | 23408 (57.58%) | 67.41 | 32.38 | |
| Yes | 17245 (42.42%) | 32.59 | 67.62 | |
| Diabetes (%) | | | | <0.0001 |
| No | 17167 (78.69%) | 88.31 | 45.93 | |
| Yes | 4650 (21.31%) | 11.69 | 54.07 | |

(*Continued*)

**Table 1.** (Continued)

| | Overall | No CKD | CKD | *p*-Value |
|---|---|---|---|---|
| | N = 40,660 | N = 33213 (81.68%) | N = 7447 (18.32%) | |
| SII (Mean±SD,1,000cells/μl) | 550.60 ± 378.90 | 544.73 ± 316.75 | 625.09 ± 461.88 | <0.0001 |

Continuous variables were expressed as mean ± SD, and P-values were calculated by the weighted linear regression model. Categorical variables are shown as percentages: p-values were calculated by weighted chi-square test. BMI, body mass index; SII, systemic immune-inflammation index; CKD: chronic kidney disease.

The red solid line represents the smoothed curve fit linking the variables, while the blue band denotes the corresponding 95% confidence interval.

In addition, we also conducted a statistical analysis of the relationship between CKD stages (G1, G2, G3a, G3b, G4, G5) and SII, as shown in S1, S2 Tables.

## Discussion

Our cross-sectional study established a positive link between SII and CKD risk. Moreover, an inflection points of 2100 (1,000 cells/μl) indicated a saturation effect in their association. Subgroup analysis indicated that the correlation between SII and CKD was more significant among hypertension or diabetes, as opposed to participants without hypertension or diabetes. These data findings imply SII is an independent risk factor for CKD when SII is below 2100 (1,000 cells/μl).

Based on literature searches conducted by the authors, our analysis is the first NHANES-based cross-sectional investigation to evaluate the association between SII and CKD in a large population with multiple circulation years. Although the relationship between SII and CKD is not yet clear, the association of SII with kidney-related disease has been demonstrated in clinical studies. A cross-sectional study discovered a beneficial relationship between SII and a high risk of kidney stones in adults under 50 years of age [27]. In a large multicenter longitudinal study, elevated SII was attached to a higher risk of overall mortality and cause-specific death among CKD patients [28]. A retrospective cohort study discovered a J-shaped connection among SII and mortality in critically ill patients with AKI [17]. Ozbek et al retrospectively analyzed 176 patients with renal cell carcinoma who underwent radical nephrectomy and found that SII was linked to increased TNM stage and a poor prognosis [18]. SII has been identified as an independent risk factor and favorable prognostic indicator for metastatic renal cell carcinoma in several studies, especially in patients with BMI $\geq$ 25 kg/m$^2$, with significant relations with both cancer-specific and overall survival [29–34]. Additionally, several retrospective and cohort research has demonstrated that SII is a hazardous indicator of contrast-induced acute kidney injury [35–41]. In addition, some studies have used SII as a predictor for the onset of acute kidney injury in diseases such as post-craniotomy in patients with spontaneous cerebral hemorrhage, post-hepatectomy in patients with hepatocellular carcinoma and severe acute pancreatitis, acute coronary syndrome and advanced chronic heart failure [42–45].

There is a strong association between inflammation and CKD based on epidemiological studies. A Randomized Controlled Trial (RCT) showed that systemic inflammation may lead to decreased physical function in patients with CKD [46]. Schanstra et al performed a proteomic analysis of 1990 participants from a large multicenter cohort and found that protein fragments linked to CKD progression were mainly derived from proteins involved in inflammation and tissue repair [47]. Additionally, another RCT demonstrated that elevated IL-6 levels were linked to an increased risk of serious adverse cardiovascular events in all stages of CKD [48]. A two-point double-blind trial showed that IL-1 trap therapy reduces systemic

**Table 2. Weighted characteristics of the study population categorized quartiles of SII.**

| | SII quartiles | | | | p-Value |
|---|---|---|---|---|---|
| | Q1 | Q2 | Q3 | Q4 | |
| | N = 10160 | N = 10165 | N = 10169 | N = 10166 | |
| Age (Mean±SD, years) | 46.88 ± 16.98 | 46.86 ± 16.65 | 47.28 ± 16.76 | 48.47 ± 17.39 | <0.0001 |
| Gender (%) | | | | | <0.0001 |
| Male | 54.22 | 49.96 | 47.43 | 41.33 | |
| Female | 45.78 | 50.04 | 52.57 | 58.67 | |
| Race (%) | | | | | 0.0291 |
| Mexican American | 20.71 | 20.19 | 21.23 | 20.79 | |
| Non-Hispanic White | 40.75 | 39.69 | 39.89 | 40.99 | |
| Non-Hispanic Black | 23.19 | 24.18 | 24.3 | 23.47 | |
| Other Hispanic | 8.22 | 8.62 | 7.55 | 7.43 | |
| Other Race | 7.13 | 7.33 | 7.03 | 7.31 | |
| Education (%) | | | | | <0.0001 |
| Less than high school | 17.19 | 15.58 | 16.14 | 16.69 | |
| High school | 22.46 | 22.21 | 24.64 | 25.18 | |
| More than high school | 60.35 | 62.21 | 59.22 | 58.13 | |
| Marital status (%) | | | | | <0.0001 |
| Never married | 18.85 | 17.06 | 17.29 | 17.18 | |
| Maried/Living with partner | 64.77 | 65.98 | 63.83 | 60.89 | |
| Widowed/Divorced/Separated | 16.38 | 16.96 | 18.88 | 21.93 | |
| Poverty ratio (%) | | | | | <0.0001 |
| 0–1.5 | 32.4 | 30.98 | 31.52 | 33.24 | |
| 1.5–3.5 | 25.76 | 24.48 | 25.02 | 26.14 | |
| >3.5 | 41.84 | 44.54 | 43.46 | 40.61 | |
| BMI (kg/m$^2$, %) | | | | | <0.0001 |
| <25 | 33.7 | 31.14 | 27.59 | 28.86 | |
| 25–29.9 | 34.59 | 34.6 | 33.29 | 30.23 | |
| ≥30 | 31.71 | 34.25 | 39.11 | 40.9 | |
| Abdominal obesity (%) | | | | | <0.0001 |
| No | 68.26 | 63.18 | 58.44 | 52.57 | |
| Yes | 31.74 | 36.82 | 41.56 | 47.43 | |
| Drinking status (%) | | | | | <0.0001 |
| Never | 12.04 | 11.18 | 10.58 | 10.99 | |
| Former | 13.11 | 12.38 | 14.36 | 16.24 | |
| Mild | 37.75 | 37.78 | 36.09 | 32.93 | |
| Moderate | 16.86 | 17.91 | 17.36 | 17.12 | |
| Heavy | 20.24 | 20.75 | 21.61 | 22.73 | |
| Smoking status (%) | | | | | <0.0001 |
| Never | 56.86 | 56.63 | 54.52 | 50.32 | |
| Former | 24.73 | 24.36 | 23.88 | 25.79 | |
| Now | 18.41 | 19.01 | 21.61 | 23.9 | |
| Hypertension (%) | | | | | <0.0001 |
| No | 64.59 | 65.22 | 62.07 | 57.92 | |
| Yes | 35.41 | 34.78 | 37.93 | 42.08 | |
| Diabetes (%) | | | | | <0.0001 |
| No | 86.29 | 84.52 | 83.03 | 78.51 | |
| Yes | 13.71 | 15.48 | 16.97 | 21.49 | |

(*Continued*)

**Table 2.** (Continued)

| | SII quartiles | | | | p-Value |
|---|---|---|---|---|---|
| | Q1 | Q2 | Q3 | Q4 | |
| | N = 10160 | N = 10165 | N = 10169 | N = 10166 | |
| CKD (%) | | | | | <0.0001 |
| No | 87.73 | 87.14 | 86.3 | 81.76 | |
| Yes | 12.27 | 12.86 | 13.7 | 18.24 | |

Continuous variables were expressed as mean ± SD, and P-values were calculated by the weighted linear regression model. Categorical variables are shown as percentages: p-values were calculated by weighted chi-square test. BMI, body mass index; SII, systemic immune-inflammation index; CKD: chronic kidney disease.

inflammation in patients with CKD [49]. One RCT showed that treatment with ticagrelor reduced the inflammatory burden through a reduction in levels of IL-1α, IL-1β, and TNFα in non-dialysis patients with CKD stage 4–5 [50]. According to our research, SII levels and CKD have a positive connection in both model 1, model 2, and model 3. A saturation effect between SII and CKD was observed in the smoothing curve and threshold analysis with an inflection point of 2100 (1000 cells/μL). On the left of the inflection point measurement, a positive association was found. On the right, however, there was no association detected, suggesting a significant threshold effect between SII and CKD. Furthermore, age, hypertension, and diabetes effects on the positive connection between SII and CKD were all statistically different. Our study found a stronger association between SII and CKD in older individuals and patients with hypertension or diabetes. According to several studies, age is an influence on the risk of CKD [51, 52]. In older individuals, increasing age did promote the progression of CKD, and their prevalence of CKD is almost 3–4 times higher than the general population, so screening for CKD in the elderly can be effective [53–56]. Many studies have already shown the importance of hypertension and diabetes in promoting the development of CKD [57–61].

There is a significant link between inflammation and CKD, although the precise mechanisms underlying this relationship are not fully known. Mendoza et al observed that levels of fibroblast growth factor 23 (FGF23) and several inflammatory markers (such as interleukin 6 (IL-6), tumor necrosis factor alpha (TNF-α), fibrinogen, and C-reactive protein (CRP)) were

**Table 3. Association between SII and CKD.**

| | Crude Model (Model 1) OR (95% CI) p-Value | Partially Adjusted Model (Model 2) OR (95% CI) p-Value | Fully Adjusted Model (Model3) OR (95% CI) p-Value |
|---|---|---|---|
| SII/100 | 1.06 (1.05, 1.06) <0.0001 | 1.05 (1.04, 1.06) <0.0001 | 1.06 (1.04, 1.07) <0.0001 |
| SII/100 quartiles | | | |
| Quartile 1 (0.02–3.36) | Reference | Reference | Reference |
| Quartile 2 (3.36–4.73) | 1.02 (0.95, 1.10) 0.5628 | 1.05 (0.97, 1.14) 0.1951 | 0.99 (0.86, 1.15) 0.9409 |
| Quartile 3 (4.73–6.69) | 1.17 (1.09, 1.26) <0.0001 | 1.21 (1.12, 1.31) <0.0001 | 1.20 (1.05, 1.38) 0.0086 |
| Quartile 4 (6.69–283.97) | 1.62 (1.51, 1.74) <0.0001 | 1.63 (1.51, 1.76) <0.0001 | 1.61 (1.41, 1.85) <0.0001 |
| p for trend | <0.0001 | <0.0001 | <0.0001 |

Model 1: no covariates were adjusted.

Model 2: age, gender, and race were adjusted.

Model 3: age, gender, race, marital status, education level, income poverty ratio, BMI, abdominal obesity, drinking status, smoking status, hypertension, and diabetes were adjusted.

| SII/100 | OR(95%CI) | | P for interaction |
|---|---|---|---|
| **Age(years)** | | | |
| 20-29 | 1.03 (0.99, 1.07) | | 0.1997 |
| 30-39 | 0.97 (0.93, 1.02) | | 0.2526 |
| 40-49 | 1.06 (1.02, 1.09) | | 0.0017 |
| 50-59 | 1.07 (1.03, 1.11) | | 0.0002 |
| 60-69 | 1.07 (1.03, 1.10) | | <0.0001 |
| 70-79 | 1.10 (1.06, 1.15) | | <0.0001 |
| 80-89 | 1.03 (0.97, 1.10) | | 0.2935 |
| **Sex** | | | |
| Male | 1.06 (1.04, 1.09) | | <0.0001 |
| Female | 1.04 (1.02, 1.06) | | <0.0001 |
| **BMI** | | | |
| 0-25 | 1.05 (1.02, 1.08) | | 0.0005 |
| 25-30 | 1.05 (1.02, 1.07) | | 0.0011 |
| >30 | 1.07 (1.05, 1.09) | | <0.0001 |
| **Abdominal obesity** | | | |
| No | 1.06 (1.04, 1.08) | | <0.0001 |
| Yes | 1.06 (1.04, 1.08) | | <0.0001 |
| **Drinking status** | | | |
| Never | 1.06 (1.02, 1.10) | | 0.0015 |
| Former | 1.07 (1.04, 1.11) | | <0.0001 |
| Mild | 1.06 (1.03, 1.09) | | <0.0001 |
| Moderate | 1.04 (1.00, 1.09) | | 0.0633 |
| Heavy | 1.05 (1.02, 1.08) | | 0.0032 |
| **Smoking status** | | | |
| Never | 1.06 (1.04, 1.08) | | <0.0001 |
| Former | 1.07 (1.04, 1.10) | | <0.0001 |
| Now | 1.04 (1.01, 1.07) | | 0.0134 |
| **Hypertension** | | | |
| No | 1.04 (1.02, 1.06) | | 0.0002 |
| Yes | 1.07 (1.05, 1.09) | | <0.0001 |
| **Diabetes** | | | |
| No | 1.04 (1.02, 1.06) | | <0.0001 |
| Yes | 1.07 (1.05, 1.10) | | <0.0001 |

0.90  1.00  1.10  1.20

**Fig 2. Subgroup analysis for the association between SII and CKD.**

increased in CKD [62]. Moreover, each unit increase in FGF23 was linked to more severe inflammation [63]. Kuo et al discovered that pro-inflammatory factors can initiate hyperoxia production in circulating mononuclear cells of CKD patients, resulting in aggravated oxidative stress, which was an essential contributor to systemic inflammation and kidney injury [64]. By increasing the release of interleukin 1 and interleukin 18, NLRP3 inflammatory vesicles have been shown to have a vital function in causing renal inflammation and fibrosis and speeding up the development of CKD [65–67]. This promotes the progression of CKD. Sjaarda et al. showed that Uromodulin and human EGF receptor 2 (HER2) are independent pathogenic mediators of CKD, and these biomarkers have potential as targets for the prevention and treatment of CKD [68]. Ruiz et al reported that targeting the transcription factor Nrf2 could reduce

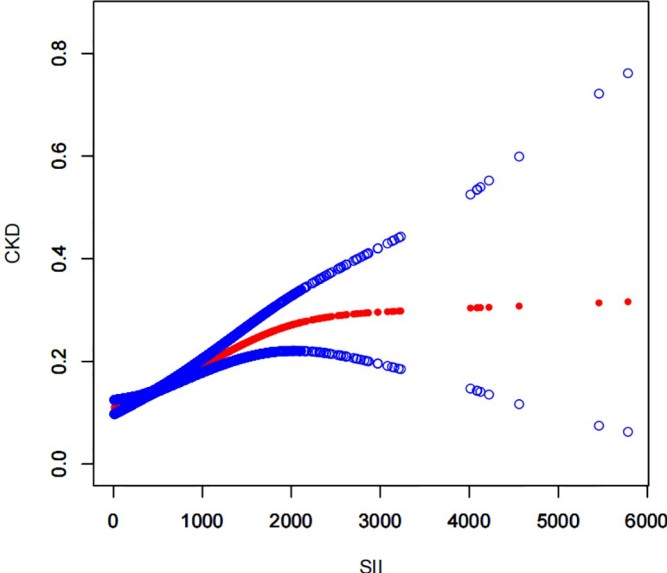

**Fig 3. Association between SII CKD.**

CKD-related oxidative stress and inflammatory responses, as this is a crucial aspect of the disease's pathogenesis [69].

Our survey has several strengths and limitations. The reliability of our study is enhanced by a large nationally representative, multi-ethnic population and appropriate covariate correction. Nonetheless, we need to consider some limitations as well. First of all, observational studies cannot determine the true causality and prospective studies are still required to support it. As this study is a cross-sectional analysis, it may not be possible to establish a definite temporality. Then, the data associated with SII were measured only once, which may underestimate the association. Similarly, since NHANES measured eGFR and uACR values only once, and the potential for acute kidney injury, the definition of CKD may not be precise enough. Furthermore, although we did our best to include appropriate confounders, we still could not exclude the effect of other possible confounders. Finally, the NHANES database limitations prevented the inclusion of drug usage as a covariate in this study, such as nonsteroidal anti-inflammatory drug use in CKD patients. Additionally, the relationship between inflammation and disease is complex and interactions were not accounted for, which may result in our findings not fully reflecting the actual situation.

**Table 4. Threshold effects of SII on CKD analyzed using linear regression models.**

|  | Adjusted OR (95% CI), *P* Value |
|---|---|
| Fitting by the standard linear model | 1.0006 (1.0004, 1.0007) <0.0001 |
| Fitting by the two-piecewise linear model |  |
| SII |  |
| Inflection point | 2100 |
| SII<2100 | 1.0007 (1.0005, 1.0008) <0.0001 |
| SII>2100 | 0.9997 (0.9991, 1.0003) 0.3151 |
| Log likelihood ratio | 0.004 |

## Conclusion

Our findings indicate a link between SII and CKD. More thorough prospective investigations are required because the results do not prove a causal link.

## Supporting information

**S1 Table. Weighted characteristics of the study population categorized by 5 stages of CKD.** Continuous variables were expressed as mean ± SD, and P-values were calculated by the weighted linear regression model. Categorical variables are shown as percentages: p-values were calculated by weighted chi-square test. BMI, body mass index; SII, systemic immune-inflammation index; CKD: chronic kidney disease.
(DOCX)

**S2 Table. Association between SII and 5 stages of CKD.** Model 1: no covariates were adjusted. Model 2: age, gender, and race were adjusted. Model 3: age, gender, race, marital status, education level, income poverty ratio, BMI, abdominal obesity, drinking status, smoking status, hypertension, and diabetes were adjusted.
(DOCX)

**S1 Checklist. STROBE statement—Checklist of items that should be included in reports of observational studies.**
(DOCX)

## Author Contributions

**Conceptualization:** Lin Li, Chengping Wen, Xiaoqin Ma, Lin Huang.

**Data curation:** Lin Li, Xiaoqin Ma, Lin Huang.

**Methodology:** Lin Li, Kunfei Chen.

**Resources:** Lin Li, Kunfei Chen.

**Supervision:** Chengping Wen, Xiaoqin Ma, Lin Huang.

**Validation:** Kunfei Chen.

**Visualization:** Lin Li.

**Writing – original draft:** Lin Li.

**Writing – review & editing:** Lin Li, Chengping Wen, Xiaoqin Ma, Lin Huang.

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
