## [Decision Letter · Decision Letter 0]

28 Jun 2023

PONE-D-23-11993Association between systemic immune-inflammation index and chronic kidney disease: a population-based studyPLOS ONE

Dear Dr. Ma,

Thank you for submitting your manuscript to PLOS ONE. After careful consideration, we feel that it has merit but does not fully meet PLOS ONE’s publication criteria as it currently stands. Therefore, we invite you to submit a revised version of the manuscript that addresses the points raised during the review process.

There are some additional corrections that I observed should be made to this manuscript in addition to those observed by the reviewers, and these are reflected in the manuscript attached.

We look forward to receiving your revised manuscript.

Kind regards,

Innocent Ijezie Chukwuonye, MBBS, FMCP(Internal Medicine)

Academic Editor

PLOS ONE

**Comments to the Author**

1. Is the manuscript technically sound, and do the data support the conclusions?

Reviewer #1: Yes

Reviewer #2: No

2. Has the statistical analysis been performed appropriately and rigorously? 

Reviewer #1: Yes

Reviewer #2: Yes

3. Have the authors made all data underlying the findings in their manuscript fully available?

Reviewer #1: Yes

Reviewer #2: Yes

4. Is the manuscript presented in an intelligible fashion and written in standard English?

Reviewer #1: Yes

Reviewer #2: Yes

5. Review Comments to the Author

Reviewer #1: The authors carried out cross-sectional study and conducted multivariate logistic regression and smoothing curves and threshold effect analyses to explore the association between SII and CKD in the adult US population. Concerns with the study design and statistical analysis are as follows:

1.Please analysis the association between CKD stage and SII if GFR data were available

2.The authors found a Inflection point in linear regression models（SI=2100). But a linear trend was found in table 3. Please provide concrete value of each quartile in Table 3

Reviewer #2: This is a novel work, written in a simple style to aid understanding. A few critical points to note:

1. Are the excluded participants systemically different form those included in the study? especially as regards age and gender distribution. This may impact on the CKD classification.

2. The definition of CKD usually requires at least 2 eGFR or uACR values obtained at least 3 months apart. It appears this study had single values for both eGFR and uACR which may affect the classification of CKD. Some of the individuals may have an AKI, thereby causing an overestimation of CKD. AKI, on its own, is an inflammatory condition which may affect SII values. This may invalidate our findings.

3. It appear the 2009 CKD-EPI equation (with race factor) was used for this analysis. This has been shown to inaccurately estimate eGFR. The more recent (and accurate) revision of the equation (DOI: 10.1056/NEJMoa2102953), should be used. The number of participants with CKD will change when the new equation is applied. This is the most important limitation of this study and should be addressed.

4. Is there a statistical trend for SII across the 5 stages of CKD? It would be good to investigate for this.

6. PLOS authors have the option to publish the peer review history of their article (what does this mean?). If published, this will include your full peer review and any attached files.

Reviewer #1: No

Reviewer #2: No

---

## [Author Response · Author response to Decision Letter 0]

22 Sep 2023

Dear Editor and Reviewers, 

We appreciate the opportunity to allow us to revise our manuscript and thanks for reviewers’ constructive comments and suggestions. Firstly, We are very sorry for the delay in revising the draft due to sudden personal problems. Once again, we would like to express my sincere apologies.

We would like to submit our revised manuscript, entitled ‘Association between systemic immune-inflammation index and chronic kidney disease: a population-based study’ for consideration for publication. In the revised manuscript, we have carefully addressed all comments and questions raised by reviewers’ point-by-point. We greatly appreciate your time and efforts to improve our manuscript for publication.

Reply to Reviewers 

Reviewer #1: 

The authors carried out cross-sectional study and conducted multivariate logistic regression and smoothing curves and threshold effect analyses to explore the association between SII and CKD in the adult US population. Concerns with the study design and statistical analysis are as follows:

1.Please analysis the association between CKD stage and SII if GFR data were available

Reply: Thank you very much for your valuable comments. We re-completed a statistical analysis of the relationship between CKD stage and SII based on GFR data. The new results can be seen in Supplementary Tables 1&2.

2.The authors found a Inflection point in linear regression models（SII=2100). But a linear trend was found in table. 

Reply: Thank you for your helpful question, which is well worth investigating. As you proposed, although our curve fitting presents atypical U-shaped or inverted U-shaped associations, inflection points are found in the threshold analysis (p for log-likelihood ratio=0.004). When SII < 2100 (1000 cells/µl), SII was positively correlated with CKD,and SII was an independent risk element for CKD. And when SII > 2100 (1000 cells/µl), there was no statistically significant association between SII and CKD.Therefore, according to this inflection point, it can be suggested that when we further explore the relationship between SII and CKD, it is more appropriate when SII < 2000(1000 cells/µl). The added description is shown in red in the paper.

3. Please provide concrete value of each quartile in Table 3.

Reply: Thank you very much for your professional guidance, which is of great help to our article. We have added specific values for each quartile based on your suggestion in Table 3. (See red in Table 3 for details)

Reviewer #2: 

This is a novel work, written in a simple style to aid understanding. A few critical points to note:

1.Are the excluded participants systemically different form those included in the study? especially as regards age and gender distribution. This may impact on the CKD classification.

Reply: Thank you very much for your valuable comments. First, excluded participants included: 1. Missing SII data (14444) 2. Missing CKD data (6028) 3. Age below 20 (19180). Since only adult participants were studied, in the age distribution, subjects younger than 20 years of age were present only in the excluded population. In addition to that, we have made a baseline table of excluded participants（See table below）based on demographic variables such as age/gender/race, showing that the mean age of excluded participants is 13.3 ± 13.4, which is different from that of included adult participants(49.77 ± 18.04) due to the excessive number of excluded participants under the age of 20 (n=19180). Age and gender remained independent risk factors for CKD (P < 0.001). In addition, among excluded participants, CKD patients were mostly white, female，which is consistent with included participants. Due to the low mean age of the excluded participants, the results were mostly those who were Less than high school, never married, more poor, BMI < 25, no abdominal obesity, no smoke, no drink, no hypertension and diabetes. Therefore, in order to achieve objective results without affecting the diagnosis and staging of CKD, participants younger than 20 years of age should be excluded due to their greater impact on the results.

Weighted characteristics of the excluded participation categorized by CKD status.(See the document for the table: Response letter.docx)

 Overall No CKD CKD p-Value

 N=39652 N= 34339(86.6%) N=5313(13.4%) 

Age (Mean ±SD, years) 13.3 ± 13.4 15.65 ± 11.71 17.01 ± 16.24 <0.0001

Gender (%) <0.0001

Male 50.8 53.3 38.01 

Female 49.2 46.7 61.99 

Race (%) 0.0132

Mexican American 16.5 16.82 16.03 

Non-Hispanic White 38.5 34.12 34.21 

Non-Hispanic Black 20.7 22.6 23.47 

Other Hispanic 9.7 10.68 9.04 

Other Race 14.5 15.77 17.25 

Education (%） <0.0001

Less than high school 80.6 81.67 85.72 

High school 8.4 8.16 5.9 

More than high school 11 10.17 8.38 

Marital status (%) <0.0001

Never married 55.1 59.9 51.4 

Maried/Living with

Partner 32.9 31.13 28.23 

Widowed/Divorced

/Separated 12 8.97 20.38 

Poverty ratio (%) 0.0018

0-1.5 44.9 44.16 40.98 

1.5-3.5 25.1 24.9 27.42 

>3.5 29.9 30.93 31.59 

BMI (kg/m², %) <0.0001

＜25 77.1 72.15 82.2 

25-29.9 12.6 15.5 8.94 

≥30 10.3 12.35 8.85 

Abdominal obesity (%) 0.1825

No 90.1 88.18 89.02 

Yes 9.9 11.82 10.98 

Drinking status (%) <0.0001

Never 22 21.76 20.23 

Former 10.5 9.32 18.6 

Mild 27 26.17 34.07 

Moderate 15.3 16.05 9.04 

Heavy 25.2 26.7 18.06 

Smoking status (%) <0.0001

Never 62.6 65.57 53.53 

Former 17.2 16.58 24.13 

Now 20.2 17.85 22.34 

Hypertension (%) <0.0001

No 92.6 94.31 89.17 

Yes 7.4 5.69 10.83 

Diabetes (%) <0.0001

No 98.5 98.49 96.16 

Yes 1.5 1.51 3.84 

SII (Mean±SD,1,000cells/µl) 437.8 ± 300.6 461.99 ± 296.11 455.29 ± 293.16 0.2894

2.The definition of CKD usually requires at least 2 eGFR or uACR values obtained at least 3 months apart. It appears this study had single values for both eGFR and uACR which may affect the classification of CKD. Some of the individuals may have an AKI, thereby causing an overestimation of CKD. AKI, on its own, is an inflammatory condition which may affect SII values. This may invalidate our findings.

Reply: Thank you for your professional advice, which has been of great help to the improvement of our article. As you pointed out, for the diagnosis of CKD, it is usually more accurate to have at least 2 eGFR or uACR values at least 3 months apart, but we are sorry that the NHANES database is a cross-sectional study with only one test value, which is also a limitation of many cohorts or databases. In addition, AKI is not yet clear in this database, so we have added it to the limitations in the discussion section.(See red in discussion for details)

3.It appear the 2009 CKD-EPI equation (with race factor) was used for this analysis. This has been shown to inaccurately estimate eGFR. The more recent (and accurate) revision of the equation (DOI: 10.1056/NEJMoa2102953), should be used. The number of participants with CKD will change when the new equation is applied. This is the most important limitation of this study and should be addressed.

Reply: Thank you very much for your valuable comments. Due to the incomplete follow-up data of the 2019-2022 NHANES database, this study only included the data of 2003-2018, and used the earlier version (2009) for formula selection. Based on your valuable comments, we have carefully studied and compared the new formula of 2021 with the formula of 2009, and found that the accuracy of the formula of 2021 is slightly lower than that of 2009 CKD-EPI creatinine formula【1-2】, so we hope to continue to use the formula of 2009. 

【1】Inker LA, Eneanya ND, Coresh J, et al. New Creatinine- and Cystatin C-Based Equations to Estimate GFR without Race. N Engl J Med. 2021;385(19):1737-1749. doi:10.1056/NEJMoa2102953 (IF: 158.5, Q1)

【2】Hsu CY, Yang W, Parikh RV, et al. Race, Genetic Ancestry, and Estimating Kidney Function in CKD. N Engl J Med. 2021;385(19):1750-1760. doi:10.1056/NEJMoa2103753 (IF: 158.5, Q1)

4. Is there a statistical trend for SII across the 5 stages of CKD? It would be good to investigate for this.

Reply: Thank you very much for your valuable comments. We re-completed a statistical analysis of the relationship between CKD stage（G1,G2,G3a,G3b,G4,G5） and SII. The new results can be seen in supplementary Tables 1&2 and in the sentences marked in red in the results section of the revised manuscript.

Finally

We marked the changes in red in the revised paper.

We have also added references, adding Supplementary Figures 1 and 2.

We have changed the manuscript according to the request using PLOS ONE' s style and added the Funding section.

These comments are all valuable and enable us to greatly improve the quality of our manuscript. We tried our best to improve the manuscript. These changes will not influence the content and framework of the paper. We appreciate for Editors/Reviewers’ warm work earnestly and hope that the corrections will meet with approval. 

Please do not hesitate to contact us with any further questions or recommendations. 

Once again, thank you very much for your comments and suggestions.

---

## [Editor Report · Decision Letter 1]

26 Sep 2023

Association between systemic immune-inflammation index and chronic kidney disease: a population-based study

PONE-D-23-11993R1

Dear Lin Huang

We’re pleased to inform you that your manuscript has been judged scientifically suitable for publication and will be formally accepted for publication once it meets all outstanding technical requirements.

Kind regards,

Innocent Ijezie Chukwuonye, MBBS, FMCP (Internal Medicine)

Academic Editor

PLOS ONE

---

## [Editor Report · Acceptance letter]

3 Oct 2023

PONE-D-23-11993R1 

Association between systemic immune-inflammation index and chronic kidney disease: a population-based study 

Dear Dr. Ma:

I'm pleased to inform you that your manuscript has been deemed suitable for publication in PLOS ONE. Congratulations! Your manuscript is now with our production department. 

Kind regards, 

on behalf of

Dr. Innocent Ijezie Chukwuonye 

Academic Editor

PLOS ONE